# PARTICLE BASED STOCHASTIC POLICY OPTIMIZATION

## ABSTRACT

Stochastic policy has been widely applied for its good property in exploration and uncertainty quantification. Modeling policy distribution by joint state-action distribution within the exponential family has enabled flexibility in exploration and learning multi-modal policies and also involved the probabilistic perspective of deep reinforcement learning (RL). The connection between probabilistic inference and RL makes it possible to leverage the advancements of probabilistic optimization tools. However, recent efforts are limited to the minimization of reverse KL divergence which is confidence-seeking and may fade the merit of a stochastic policy. To leverage the full potential of stochastic policy and provide more flexible property, there is a strong motivation to consider different update rules during policy optimization. In this paper, we propose a particle-based probabilistic policy optimization framework, ParPI , which enables the usage of a broad family of divergence or distances, such as $f$-divergences and the Wasserstein distance which could serve better probabilistic behavior of the learned stochastic policy. Experiments in both online and offline settings demonstrate the effectiveness of the proposed algorithm as well as the characteristics of different discrepancy measures for policy optimization.

## 1 INTRODUCTION

Deep reinforcement learning (DRL) leverages the power of neural-network function approximators and has shown great promise in a diverse field, such as control (Lillicrap et al., 2015; Andrychowicz et al., 2020), robotics (Haarnoja et al., 2017; Gu et al., 2017; Plappert et al., 2018). Classic views on the notion of optimality in reinforcement learning state that an optimal policy can be deterministic (Sutton & Barto, 2011). However, recent studies by Daniel et al. (2012); Ziebart (2010) demonstrate that deterministic policy may suffer from several drawbacks, *e.g.*, the multi-modal behaviors have significant applications in real robotic control tasks (Daniel et al., 2012) and inflexibility on exploration (Ostrovski et al., 2017). Particularly in offline RL, the deterministic policy lacks the structure to manage risk towards uncertainty dynamics. The branch of probabilistic RL is then introduced where the optimal solution is stochastic. Probabilistic RL not only enables the stochasticity of the learned policy but also provides a different perspective to reveal the connection between reinforcement learning and probabilistic inference (Todorov, 2008), where the optimization procedure of the policy could be considered a statistical divergence minimization procedure towards distribution in the state-action or trajectory space, which is implied by the corresponding optimal RL policy.

Besides the different definitions of optimality and above mentioned priorities, another appealing property of probabilistic reinforcement learning lies in that advanced techniques in approximate inference can then be applied in the optimization for RL. There are two widely applied frameworks, *i.e.*, the pseudo-likelihood framework and maximum entropy (MERL) framework. The pseudo-likelihood defines the optimality by considering a specified positive and bounded function of reward as the density function. By regressing to the reweighted samples, the pseudo-likelihood based methods not only is not an explicit probabilistic model but also suffers from high variance and poor risk handling (Levine, 2018). MERL is a prominent choice for policy improvement for stochastic policy which tends to maximize the expected reward and the expected conditional entropy. It avoids the drawbacks of pseudo-likelihood that it suffers from high variance and poor risk handling ability (Levine, 2018). In practice, the MERL is generally implemented with the reverse KL minimization which promotes

the mode-seeking behavior and could fade the merits of a stochastic policy. For example, in offline setting, the mode-seeking policy will result in an overly-optimistic estimation of $Q$-function (Kumar et al., 2020) which could be problematic when generalizing in the test scenario.

In this work, we explore the possibility of using a broad class of distribution discrepancies for stochastic policy improvement, and find it is possible by leveraging particle-based methods. By providing better distribution matching methods, our proposed framework, ParPI , enjoys the power of probabilistic RL. Moreover, ParPI is naturally compatible with the offline RL setting. And we show that with specific discrepancy measure selected, ParPI could conjoin the benefit of different policy constraint methods. We firstly demonstrate that using a broader class of distribution discrepancies achieves significant improvements in stabilizing the policy optimization and achieving the state-of-the-art expected returns on high-dimensional and complex continuous control tasks (Todorov et al., 2012). Besides, we conduct extensive experiments in the offline setup (Yu et al., 2020) to show that the ParPI could be more defensive for optimization thus results in more generalizable optimal policy.

## 2 PRELIMINARIES

### 2.1 PROBABILISTIC INFERENCE FRAMEWORK FOR RL

The reinforcement learning (RL) problem can be modeled as the following infinite-horizon Markov Decision Process (MDP), which is implied by the tuple $\langle \mathcal{S}, \mathcal{A}, r, p, p_0, \gamma \rangle$. We consider the common cases where the state space $\mathcal{S}$ and action space $\mathcal{A}$ are assumed to be continuous. $p(s_{t+1}|s_t, a_t)$ refers to the transition probability, *i.e.* the probability density of next state $s_{t+1}$ given the current state $s_t$ and action $a_t$. when an agent chooses an action $a$ in state $s$, the environment emits a reward $r(s, a) : \mathcal{S} \times \mathcal{A} \to [r_{\min}, r_{\max}]$ during the transition. The policy distribution $\pi(a|s)$ is presented as a distribution over the action space conditioned on any state. The distribution of state-action pairs when navigating the environment with policy $\pi$ could be defined as $\rho_\pi(s, a)$, which is also referred to as occupancy measure (Ho & Ermon, 2016), and the marginal distribution on the state space is $\rho_\pi(s)$. With a predefined MDP, the notation trajectory $\tau_\pi$ refers to the following Markov sequence $\tau_\pi = \{(s_0, a_0), \cdots (s_T, a_T)\}$ generated by interacting with environment under policy $\pi$. The discount factor is denoted as $\gamma$. Standard reinforcement learning seeks to find the optimal policy that could maximize the expected return, and the optimal solution could be deterministic (Sutton & Barto, 2011). However, considering the exploration (Schulman et al., 2015), robustness (Ziebart, 2010) and multi-modal objectives (Daniel et al., 2012) in real-world application scenarios, a stochastic/multi-modal policy is desired. To address this desideratum, optimal maximum entropy policy (Ziebart et al., 2008; Haarnoja et al., 2017) is defined as:

$$\pi^* := \underset{\pi}{\arg\max} \sum_t \mathbb{E}_{\rho_\pi(s_t, a_t)}[r(s_t, a_t) + \alpha \mathbb{H}[\pi(\cdot|s_t)]].$$

and the soft Bellman optimality equation (Ziebart, 2010) is defined as:

$$Q^*(s_t, a_t) = r_t + \gamma \mathbb{E}_{s_{t+1}}[V^*(s_{t+1})], \quad (1)$$

which gives the soft Bellman operator $\mathcal{T}^\pi Q^\pi(\cdot) = Q^\pi(\cdot)$ and $\mathcal{T}^\pi \cdot := r(h) + \gamma \mathbb{E}_{h' \sim p(s'|h)\pi(a'|s')}[\cdot]$ and also referred to as Maximum Entropy RL(MERL). Ziebart (2010); Haarnoja et al. (2017) and Haarnoja et al. (2018) show that the target of the policy update is given by the Q-function in an energy-based form, $\pi(a|s) \propto \exp(-Q(s, a)/\alpha)$. Due to the intractability in estimating the normalizing constant $Z(s) = \int \exp(-Q(s, a)/\alpha) \mathrm{d}a$ for continuous or large discrete action space, the energy-based form poses difficulties in utilizing the policy such as sampling or density estimation, so various methods use a standalone parametric policy model $\pi_\phi(a|s)$ to distill from the energy function, and the MERL task is turned into iteratively updating the value model $Q_\theta$ and the policy model $\pi_\phi(a|s)$.

## 3 PARTICLE-BASED POLICY IMPROVEMENT

### 3.1 DISTRIBUTION MATCHING FOR POLICY IMPROVEMENT

We aim to update a policy model to match a better target $p(a|s)$, which is demonstrated by particles. We now formalize it as a distribution matching task under a general distribution metric/discrepancy $\mathrm{D}[\cdot\|\cdot]$. The first problem is that the policy model parameterizes a conditional distribution $\pi_\phi(a|s)$, so we need to match the target $p(a|s)$ by minimizing $\mathrm{D}(\pi_\phi(a|s), p(a|s))$, however, it is intractable for

any given $s$. To make the minimization practical, we note that in almost all modern policy update tasks, we also have an associated state trajectory, *e.g.*, experiences in a replay buffer in off-policy RL, or simulated trajectories in inverse RL (Finn et al., 2016) and demonstrative trajectories in imitation learning (Ho & Ermon, 2016). This state trajectory demonstrates a marginal distribution on the state, $\rho(s)$, which is widely assumed to be stationary among different policies within the same MDP (Tsitsiklis & Van Roy, 1996; Haarnoja et al., 2017; 2018). Hence we instead minimize $D(\rho(s)\pi_\phi(a|s), \rho(s)p(a|s))$. When the joint distributions match, the conditionals also match. This idea has been exploited in other fields, such as expectation propagation algorithms (Minka, 2001) for Bayesian inference. And we further provide discussions in the Appendix F. Along with the state trajectory, we also have samples/particles from the augmented target, $\rho(s)p(a|s)$. Particularly for existing MERL methods, they minimize the reverse KL divergence $D_{KL}(\pi_\phi(\cdot|s), p(\cdot|s))$ averaged over a state distribution $\rho(s)$ which can be implemented by conducting sampling from the replay buffer: $\mathbb{E}_{\rho(s)}[D_{KL}(\pi_\phi(\cdot|s), p(\cdot|s)]$. This objective can be reformulated as $D_{KL}(\rho(s)\pi_\phi(a|s), \rho(s)p(a|s))$, corresponding to our proposed method with $D$ instantiated as the reverse KL.

As stressed, this framework allows matching the distribution by minimizing more powerful metrics/discrepancies that do not have a particular propensity, like the reverse KL. To balance utility and implementability, we consider the $f$-divergences $D_F$ and the Wasserstein distance (WD) $D_W$ as two instances of $D[\cdot\|\cdot]$ of the proposed framework.

**Minimizing f-divergences for policy updates.** The $f$-divergence is defined as $D_F(q, p) := \mathbb{E}_p[F(q/p)]$, where $F$ is a real-valued convex function with $F(1) = 0$. Although the forward and reverse KL are also f-divergence instances, we do not consider them here since they would introduce the respective propensity of the learned model. Nowozin et al. (2016) show a useful lower bound of an f-divergence that benefits practical optimization:

$$D_F(q, p) \geqslant \sup_f \left( \tilde{D}_F(q, p; f) := \mathbb{E}_q[f(\cdot)] - \mathbb{E}_p[F^*(f(\cdot))] \right) \qquad (2)$$

where $F^*(t) := \sup_x\{xt - F(x)\}$ is the Fenchel conjugate of $F$. If the parametric model is defined via a reparameterization $h \sim q_\phi(h) : h = g_\phi(\epsilon)$ with $\epsilon$ following a fixed and easy-to-sample distribution $q(\epsilon)$ like the standard Gaussian, the first term can also be estimated as $\mathbb{E}_{q(\epsilon)}[f(g_\phi(\epsilon))]$. By also parameterizing $f$ in some function class, the supremum can be estimated after optimizing over $f$, which also tightens the lower bound. Thus minimizing $\tilde{D}_F$ well serves as minimizing $D_F$.

**Minimizing WD for policy updates.** The WD is defined as the minimal cost of transferring from one distribution to the other by a probabilistic transferring plan. It is shown to have an optimization utility even when the two distributions do not have overlapping support (Arjovsky et al., 2017). Its formulation under Kantorovich-Rubinstein duality makes it convenient to optimize:

$$D_W(q, p) = \sup_{\|f\|_L \leqslant 1} \mathbb{E}_q[f] - \mathbb{E}_p[f] \qquad (3)$$

For distributions $q, p$ with a bounded expectation on a Polish metric space (Villani, 2008, Particular Case 5.16) (e.g., the common Euclidean space), where $\|\cdot\|_L$ denotes the Lipschitz constant of a function. To enforce the Lipschitz constant constraint, various implementations are proposed, e.g. parameter clipping (Arjovsky et al., 2017), gradient penalty (Gulrajani et al., 2017) and spectral norm / hinge loss regularizations (Miyato et al., 2018).

**Framework 1.** *The Particle-based Policy Improvement (ParPI) framework, $\min_\phi D(\rho(s)\pi_\phi(a|s), \rho(s)p(a|s))$ can be used for the sub-task of updating policy in RL, as long as the improved policy can be demonstrated by particles. The process is shown in Fig. 1.*

**Remark 1.** *The ParPI framework can be applied to policy update in SQL (Ziebart, 2010; Haarnoja et al., 2017) and SAC (Haarnoja et al., 2018). In SQL, once the optimal Q-function $Q^*$ is achieved, the optimal policy is shown to be $\pi^*(a|s) = \exp(Q^*(s, a)/\alpha)/Z^*(s)$ where $Z^*(s)$ is a normalizing constant. In SAC, it is shown that $\pi'(a|s) := \exp(Q^\pi(s, a)/\alpha)/Z(s)$ is a better policy than $\pi$ where $Q^\pi$ is the Q-function under policy $\pi$. In both cases, the policy update target is given in an energy-based form, whose particles can be drawn using MCMC algorithms.*

## 3.2 ParPI for MERL

The particle-based policy improvement (ParPI) framework can be applied to any RL task where a state-action trajectory demonstrates a better policy. We now show in detail how it helps to update policy in the MERL framework. As introduced, MERL (Ziebart et al., 2008; Ziebart, 2010; Haarnoja et al., 2017; Liu et al., 2017; Haarnoja et al., 2018; Zhang et al., 2018) defines the optimal policy which also considers the entropy to cover multiple optimalities and encourage exploration. Based on the definition, a policy update target is given in the form $p(a|s) \propto \exp(Q(s,a)/\alpha)$, where $Q(s,a)$ is either the optimal Q-function or the Q-function of the current policy, which can be estimated from simulated trajectories. These methods update the policy model by minimizing the reverse KL $D_{KL}(\rho(s)\pi_\phi(a|s), \rho(s)p(a|s))$. The merit of it is that estimating the troublesome normalizing constant is not required in optimization, which is also widely

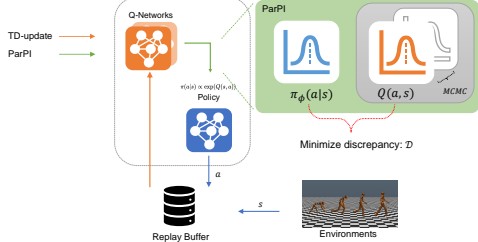

Figure 1: The Particle-based Policy Improvement (ParPI) framework. The current Q-function provides a better policy in an energy-based form, whose particles can be drawn by an MCMC algorithm. The policy model is updated to match the better policy by minimizing a broader class of distribution metrics using the particles.

exploited in variational inference. However, it has been pointed out (Huszár, 2015; Theis et al., 2016) that minimizing the reverse KL promotes the mode-seeking behavior in finite optimization steps, making $\rho(s)\pi_\phi(a|s)$ concentrate to one mode of the target $\rho(s)p(a|s)$. This is because a substantial penalty can be incurred when $\rho(s)\pi_\phi(a|s)$ puts more mass to where $\rho(s)p(a|s)$ is dilute, but a regular loss in the other way. The behavior is also observed in practice, and there are attempts to ameliorate it in other fields, like variational inference (Hernández-Lobato et al., 2016; Li & Gal, 2017) As the goal of MERL is to capture the diversity and multi-modality of policy, such behavior would weaken the spirit of MERL.

The ParPI framework allows MERL to benefit from more powerful metrics/discrepancies, as long as the trajectory samples are available. To draw from the target policy, we employ the Langevin dynamics, which only requires an unnormalized density function of the target distribution.

**Sampling with Langevin Dynamics.** To minimize the general metrics/discrepancies above, samples from the target distribution $\rho_\pi(s)p(a|s)$ are required. Fortunately, this can be done by subsequently sampling from $\rho_\pi(s)$ and $p(a|s)$, and both can be implemented. Samples from $\rho_\pi(s)$ can be drawn by simulating the $(s,a)$ trajectory following the current policy $\pi(a|s)$ and environment transition. Sampling from $p(a|s)$ can be done by running dynamics-based MCMC algorithms, which only require the unnormalized density, i.e. $\exp(Q(s,a)/\alpha)$. From various particle-based inference algorithms, we employ the Langevin dynamics (Roberts et al., 1996; Roberts & Stramer, 2002; Welling & Teh, 2011) due to the low computational cost, whose transition is given by:

$$a^{(i+1)} = a^{(i)} + \varepsilon\nabla\log p(a^{(i)}|s) + \mathcal{N}(0, 2\varepsilon I) = a^{(i)} + (\varepsilon/\alpha)\nabla_a Q(a^{(i)}, s) + \mathcal{N}(0, 2\varepsilon I) \quad (4)$$

It has been shown to achieve $\delta$ precision in total variance (Dalalyan, 2017), Wasserstein distance (Durmus & Moulines, 2016) and KL divergence (Cheng & Bartlett, 2017) in $O(1/\delta^2)$ steps for strongly log-concave densities. Note that although we can use MCMC to draw samples from the policy target, the sampling is slow in deploying the policy, where we need to run Markov chains for every state along the trajectory, so explicitly modeling policy $\pi_\phi(a|s)$ is still necessary.

**Value Function Update.** The proposed policy improvement rule described above only requires a model for the state-action value function $Q_\theta(s,a)$, so it can be applied with any value function update method. The soft-Q learning method (Haarnoja et al., 2017) aims to update the value function towards optimal by a temporal difference implementation based on the soft Bellman optimality equation 1:

$$J_Q(\theta) = \mathbb{E}_{(\mathbf{s}_t,\mathbf{a}_t)\sim\mathcal{D}}\left[\frac{1}{2}\left(Q_\theta(\mathbf{s}_t,\mathbf{a}_t) - r(\mathbf{s}_t,\mathbf{a}_t) + \log\mathbb{E}_{\mathbf{s}_{t+1}\sim p(\mathbf{s}_{t+1}|\mathbf{s}_t,\mathbf{a}_t)}[\exp(V(\mathbf{s}_{t+1}))]\right)^2\right] \quad (5)$$

The soft actor-critic (SAC) method (Haarnoja et al., 2018) instead updates the state-value function model towards the state-action value of the current policy which gives a better policy target as following:

$$J_V(\psi) = \mathbb{E}_{\mathbf{s}_t \sim \rho(s)} \left[ \frac{1}{2} \left( V_\psi(\mathbf{s}_t) - \mathbb{E}_{\mathbf{a}_t \sim \pi_\phi} \left[ Q_\theta(\mathbf{s}_t, \mathbf{a}_t) - \log \pi_\phi(\mathbf{a}_t|\mathbf{s}_t) \right] \right)^2 \right]$$

$$J_Q(\theta) = \mathbb{E}_{(\mathbf{s}_t, \mathbf{a}_t) \sim \mathcal{D}} \left[ \frac{1}{2} \left( Q_\theta(\mathbf{s}_t, \mathbf{a}_t) - r(\mathbf{s}_t, \mathbf{a}_t) + \gamma \mathbb{E}_{\mathbf{s}_{t+1} \sim p(\mathbf{s}_{t+1}|\mathbf{s}_t, \mathbf{a}_t)} \left[ V_{\bar{\psi}}(\mathbf{s}_{t+1}) \right] \right)^2 \right]$$

(6)

And we provide convergence analysis of ParPI in Appendix E.

## 3.3 ParPI for Offline RL

Another important advantage of ParPI lies in that it is naturally compatible with the offline RL setting. In the offline RL setting, we only have access to some static dataset, *i.e.* $\mathcal{D} = \{s, a, r, s_0\}$, which is usually collected by the behavior policy $\pi_B$. And during the learning procedure, the algorithm could not interact further with the environment. Thus the key challenge for offline RL algorithms is to generalize beyond the state and action support of the offline data. There are two mainstream offline RL algorithms: the value function regularization methods and the policy constrained methods. The value function regularization methods generally refers to the methods which conservatively regularized the estimation of value function in either model-free or model based fashion, *e.g.* MOPO (Yu et al., 2020) and CQL (Kumar et al., 2020). Such methods do not have requirements on the policy improvement procedure, thus we could just directly change the vanilla policy optimization algorithm, *e.g.* entropy regularized update (Haarnoja et al., 2018), into ParPI as illustrated in Remark 1.

We mainly discuss the relationship between the policy constrained based offline RL algorithm and ParPI . The policy constrained offline RL algorithm tends to stabilize the training by adding constraints into the policy improvement procedure. Without loss of generality, the constrained policy objective could be then formalized as:

$$\pi_\phi = \arg\max_{\pi_\phi} \mathbb{E}_{s \sim \mathcal{D}, a \sim \pi_\phi(a|s)}[Q(s, a)] \text{ s.t. } D\left(\pi_\phi(a|s), \pi_\beta(a|s)\right) \leq \varepsilon \quad (7)$$

Here $D(\cdot, \cdot)$ stands for some discrepancy measure between two distributions, and $\pi_\beta$ is the behavior policy. Two main types of policy constraints are distributional constraints and support constraint. The distribution constraints (Nair et al., 2020; Wang et al., 2020; Peng et al., 2019; Wu et al., 2019) are generally more strict, which could provide a stable supervision signal yet limit the search space for the learned policy. And they are usually implemented by the minimization of density-based divergence, *e.g.* KL divergence and $f$-divergence. In comparison, the support constraints are relatively loose and only require the support of learned distribution to be equal to the behavior distribution. The support constraint enlarges the search space while it could bring unstable optimization when the support distance is large. Correspondingly, the support constraint compares samples regardless of density which is in line with the Integral Probability Metric (IPM), *e.g.* Maximum Mean Discrepancy and Wasserstein distance.

We then show that within a specific choose of divergence in ParPI , we could conjoin the benefit of both the distribution constraints and support constraint. With the $D(\cdot, \cdot)$ as reverse KL divergence, the Lagrangian duality of Eq. 7 could be formulated as: $\pi_\phi = \arg\max_{\pi_\phi} \mathbb{E}_{s \sim \mathcal{D}, a \sim \pi_\phi(a|s)}[Q(s, a)] + \alpha D_{\mathrm{KL}}\left(\pi_\phi(a|s), \pi_\beta(a|s)\right)$ where $\alpha$ is the Lagrangian multiplier. The above formulation has the closed-form solution $\pi_{\phi^*}(a|s) \propto \pi_\beta(a|s) \exp(Q(s, a)/\alpha)$ and we could then use ParPI to optimize $\pi_\phi$ towards $\pi_{\phi^*}$. Particularly, the $\nabla \log p(a^{(i)}|s)$ in Eq. 4 is $\nabla \pi_\beta(a|s) \exp(Q(s, a)/\alpha)$ in this case and the initial state of langevin dynamics is set as the behavior policy $\pi_\beta$. And the Wasserstein distance is used in particle optimization. Then we have the following fact:

**Remark 2.** *With limited steps of langevin dynamics, ParPI is approximately solving the optimization problem:*

$$\phi_{k+1} = \arg\min_\phi \mathrm{KL}(\pi_\phi(a|s) \| \frac{\pi_\beta(a|s) \exp(Q(s, a)/\alpha)}{Z}) + \alpha W_2^2\left(\pi_\phi, \pi_\beta\right) + \gamma W_2^2\left(\pi_\phi, \pi_{\phi_k}\right) \quad (8)$$

*Here $Z$ stands for the normalizing constant and the $W_2^2(\cdot, \cdot)$ is the $W_2$ distance.*

We leave the formal discussion of Remark. 2 in the Appendix D. The first two terms in Eq. 8 correspond to the distribution constraint and the support constraint. And the last term is a specific property introduced by ParPI which regularizes the update of policy to be close to the current policy. The intuition ensembles the trust region methods such as TRPO (Schulman et al., 2015) and PPO (Schulman et al., 2017) while the regularization of ParPI is conducted according to Wasserstein distance instead of KL divergence.

## 3.4 PRACTICAL OPTIMIZATION WITH PARPI

**Estimating and Minimizing the Discrepancy Measure** The occupancy measure from the current policy can be sampled by firstly sampling states from replay buffer and take action under the current policy $\pi_\phi(a|s)$ on each state, we carefully overload the notation and refer to the corresponding $(s, a)$ distribution as $\rho_\phi$. As introduced in Eq.4, the occupancy measure implied by the $Q_\theta$ can approximately be retained by finite steps in Langevin Dynamics. In practice, we use the samples from $\rho_\phi$ as the initial state, and the corresponding distribution is referred to as $\rho_Q$.

With the samples from $\rho_\phi$ and $\rho_Q$ available, we parameterize the discriminator(critic) as $f_\omega$ to compute the objective in Eq.2 and Eq.3. More specifically, the Lipschitz constraint in minimizing Wasserstein Distance(WD) cases is imposed by spectral normalization following (Miyato et al., 2018). The weight matrix $W$ in the discriminator network is regularized as $\bar{W}_{\text{SN}} := \frac{1}{\sigma(W)}W$ where $\sigma(W)$ denotes the largest singular value of $W$. Empirically, the amortization optimization procedure of finding the appropriate $\phi$ of policy model can lead to an intractable problem due to the complexity of the space $\mathcal{S} \times \mathcal{A}$ which leashes the very purpose of function approximation. To counter this obstacle, following the recent success in apprenticeship learning and imitation learning (Ho & Ermon, 2016), we add an entropy regularizer to the objective of the policy(actor) model. Without loss of generality, we reparameterize the policy model $(\pi_\phi(a_t|s_t))$ as a neural network transformation:

$$a_t = \mu_\phi(s_t) + \sigma_\phi(s_t)\epsilon_t, \epsilon_t \sim p(\epsilon),$$

so $\pi_\phi(a|s) = p_\epsilon\left(\frac{a-\mu_\phi(s)}{\sigma_\phi(s)}\right)/\sigma_\phi(s)$. Before each policy update, we first update the discriminative(critic) function $f_\omega$ for estimating $\tilde{D}_F$ or $\text{D}_\text{W}$ using particles of the target policy $\{(a'_t, s_t)\}_t$, by maximizing

$$J_f(\omega) := \begin{cases} \mathbb{E}_{\rho(s)\pi_\phi(a|s)}[\log \text{Sigm}(f_\omega(s, a))] - \mathbb{E}_{\rho(s)p(a|s)}[\log \text{Sigm}(1 - f_\omega(s, a))] \\ \approx \log \text{Sigm}\big(f_\omega(s_t, \mu_\phi(s_t) + \sigma_\phi(s_t)\epsilon_t)\big) - \log \text{Sigm}(1 - f_\omega(s_t, a'_t)), & \text{for JSD,} \\ \mathbb{E}_{\rho(s)\pi_\phi(a|s)}[f_\omega(s, a)] - \mathbb{E}_{\rho(s)p(a|s)}[f_\omega(s, a)] \\ \approx f_\omega(s_t, \mu_\phi(s_t) + \sigma_\phi(s_t)\epsilon_t) - f_\omega(s_t, a'_t), & \text{for WD,} \end{cases} \quad (9)$$

take Jensen-Shannon Divergence(JSD) as an example of f-divergence and Sigm is the sigmoid function. After updating $\omega$, we update the policy model by minimizing $J_\pi(\phi) := -\mathbb{E}_\rho(s)[\mathbb{H}[\pi_\phi(\cdot|s)]] + D(\rho(s)\pi_\phi(a|s), \rho(s)p(a|s))$ as the final objective.

**Squashing Correction and Sampling in Latent Space** Following the default setting in (Haarnoja et al., 2018), we model the action distribution with an unbounded Gaussian. As the experiment settings always limit the action space in a finite interval, we apply $\tanh$ on the samples from Gaussian. However, if we conduct the Langevin Dynamics(Eq. 4) in the bounded support, *e.g.* $(-1, 1)$, we empirically find the method is susceptible to the selection of noise level $\epsilon$ and some numerical issues could raise due to that the produced samples could be out of the boundary. To alleviate the above issues, we do the MCMC steps in the raw action space before squashing. Here, we slightly overload the notation by denoting the random variable of the raw output as $u$ which has infinite support and the variable of corrected action is $a = \tanh(u)$. With the stationary distribution in the action space as $\exp(Q(s, a)/\alpha)$, we could get the corresponding stationary distribution in $u$ space by applying change of variable formula. The score $\nabla_u \log p(u|s)$ used in Langevin Dynamics, *i.e.* Eq. 4, is:

$$\nabla_u \log p(u|s) = \nabla_u(Q(s, \tanh(u))/\alpha + 2\sum_{i=1}^{D} \nabla_u \log(1 - (\tanh(u_i)))$$

where $u_i$ denotes the $i$-th element of $u$.

## 3.5 DISCUSSION AND RELATED WORKS

The final objective is related to the regularized variants of apprenticeship learning algorithms in imitation learning (Ho et al., 2016; Syed et al., 2008). While in our setting, the corresponding target occupancy measure is defined implicitly also changes along with the optimization of Q-function

---

**Algorithm 1** ParPI:Particle-Based Policy Improvement

---

1: **Input:** Replay buffer $\mathcal{D}$. Policy model $\pi_\phi(a|s)$, parameterized Q-function $Q_\theta$, state function $V_\psi$ and discriminator(critic) $f_\tau$.
2: Set the step size $\epsilon$, the length of MCMC steps $K$ and the total iterations $T$.
3: **while** not converged **do**
4:      Sample a batch of triple $\{(s,a,r)_t\}_{t=1}^m$ from the replay buffer $\mathcal{D}$.
5:      Conducting update on the $Q_\theta$:
6:      $\theta_i \leftarrow \theta_i - \lambda_Q \nabla_{\theta_i} J_Q(\theta_i)$ for i $\in \{1,2\}$                          Eq. 5 or Eq. 6
7:      $\psi \leftarrow \psi - \lambda_V \nabla_\psi J_V(\psi)$                                       Eq. 6 # ParPI + SAC
8:      Sample a batch of state $\{s_t\}_{t=1}^m$ from $\mathcal{D}$.
9:      Draw action under current policy and get the state-action pairs $\{s_t, a_{\pi_\phi}\}_{t=1}^m$
10:      Fixing the state, and conduct Langevin dynamics with $Q_\phi$ following equation 4 to acquire the updated state-action pairs $\{s_t, a_{\hat{\pi}_\phi}\}_{t=1}^m$.
11:      $\omega \leftarrow \omega - \lambda_f \nabla_\omega J_f(\omega)$                                          Eq. 9.
12:      $\phi \leftarrow \phi - \lambda_\pi \nabla_\phi J_\pi(\phi)$.
13: **end while**

---

which indicates a more difficult optimization problem. Liu et al. (2017) consider a particle-based policy update, but rather than the target policy, their particles represent the posterior of the policy model parameter, and the update rule is also based on minimizing the reverse KL. Zhang et al. (2018) also propose a particle-based method for policy update by utilizing the minimal movement discretization of the Wasserstein gradient flow to minimize, but still the reverse KL. Many recent methods learn energy-based models especially with deep models (Du & Mordatch, 2019; Song & Ermon, 2019; Li et al., 2019a), while we instead learn a parametric model to approximate a distribution defined by an energy function for efficient prediction. We note that there are other methods to do so like the amortized MCMC methods (Li et al., 2017; 2019b; Feng et al., 2017), but the stationality-oriented objective may not be efficient for directly matching the target, and we cover more scenarios where only the particles are available. Our method is also related to the policy constraint methods in the offline RL scenario. Wu et al. (2019) and Kumar et al. (2019) proposed to constrained the learned policy with $f$-divergence or IPM, while their methods differ from us in the fact that the target occupancy could be directly acquired from the static dataset and the divergence minimization term is only served as a regularizer.

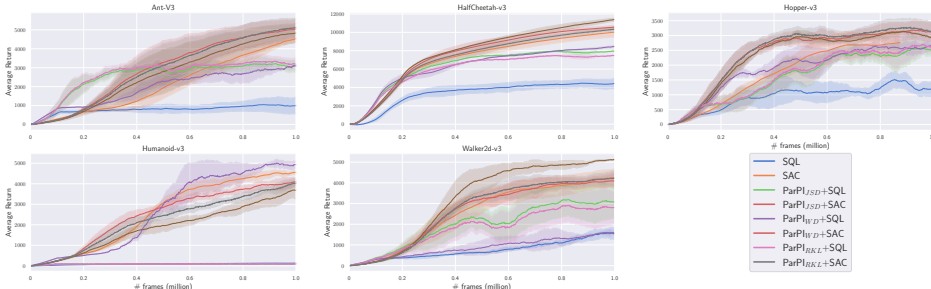

Figure 2: Training curves on continuous control benchmarks (Todorov et al., 2012). Our ParPI framework improves the performance of base RL algorithms SQL and SAC, and achieves the best results consistently across all tasks. We average the return over the past 100 episodes, where the solid lines indicate the mean and shaded areas indicate the standard deviation.

## 4 EXPERIMENTS

We conduct extensive experiments on both online scenarios and offline scenarios. Besides, we delve deeply into how the component would affect the performance of ParPI on task HalfCheetah-v3.

### 4.1 ONLINE SCENARIOS

We choose five well-known representative benchmark tasks with continuous control (Ant, Hopper, Humanoid, HalfCheetah and Walker2d) from Mujoco (Todorov et al., 2012). The method is compared

to SQL (Haarnoja et al., 2017) and SAC (Haarnoja et al., 2018), which represents the SoTA model-free off-policy methods for continuous control tasks. We choose Reverse KL divergence(RKL) and Jensen-Shannon Divergence(JSD) out of the $f$-divergence family. For the Wasserstein distance, we adopt the hinge loss (Miyato et al., 2018). And the Lipschitz constraint is enforced through gradient penalty (Gulrajani et al., 2017) and spectral normalization (Miyato et al., 2018). The proposed method is robust, We keep all of ParPI 's hyper-parameter constant across all tasks in all domains for simplicity. For hyper-parameters and additional implementation details, please refer to Appendix G. All our experiments are conducted on machines equipped with Nvidia P100 GPUs using five different random seeds.

As shown in Figure 2,we see that ParPI yields significant improvements in both performance and sample efficiency across a wide range of environments. Comparing learning curves of different methods, ParPI improves both SQL and SAC by a significant boost with almost no hyper-parameter tuning effort. One observation is that the ParPI with SQL shows more improvement then with SAC. The reason lies in the fact that the policy implied by the Q function in SAC is also shifted along with the training, which could be the unstable element for the optimization of ParPI . And such situation could be alleviated by optimizing the Q network for more steps than the policy network in each update. Here we only involve the same training configurations for fair comparison. Interestingly, we found that the Reverse KL divergence with ParPI has still consistently outperformed the original SQL and SAC which optimized the reverse KL divergence in the primal form. This could be due to that the intermediate target distribution constructed by MCMC smooths the density ratio estimation procedure (Rhodes et al., 2020) and favors the dual form optimization. The fact further justifies the effectiveness of proposed framework.

### 4.2 OFFLINE SCENARIOS

To further demonstrate the efficacy of ParPI , we then experiment with the D4RL offline reinforcement learning benchmark (Fu et al., 2020), including three environments(hopper, walker2d, and halfchee-tah), which consist of five different logged data-sets types (random, medium, medium-replay, expert, and medium-expert), in a total of 15 sub-tasks. These datasets are generated by an agent using SAC in rlkit (Pong, 2020), with each dataset containing 1 million time-steps of environment interaction. As we discussed in Sec. 3.3, we study the performance of ParPI from the combination with both value function regularized methods and policy constraint methods. Particularly, we implement ParPI on top of MOPO (Yu et al., 2020) and BRAC (Wu et al., 2019)(see Appendix H), and the discrepancy measure in ParPI is set as wasserstein distance. We follow the same schema as the D4RL paper (Fu et al., 2020) to calculate the normalized return. And the result of 15 sub-tasks is presented in Tabel 1. Compared to the baseline, our approach achieves better performance in most environments, showing the power of ParPI .

### 4.3 COMPONENT ANALYSIS AND COMPUTATIONAL EFFICIENCY

We conduct ablation study of different discrepancy measures and MCMC parameters on the task *HalfCheetah-v3* with the same settings as $\mathrm{ParPI} + \mathrm{SQL}$ (see 1).

**Discrepancy Measure** We demonstrate the characteristics of different distribution metrics for policy optimization including: JSD, original WD and the hinge loss. Here we retain all the hyper-parameters of our method and the choice of the metric is the only difference. Interesting results can be observed from Figure 3: while the entropy of learned policy remains almost the same for different metrics, the standard deviation varies. The large standard deviation indicates that the learned policy tends to be more certain on frequently observed state regions and does not mislead in other regions. According to the empirical performance, we provide the following guidelines for the future usage of ParPI : in general the Wasserstein distance could be a more desirable discrepancy measure and it also achieves the best performance in most tasks; Jensen-Shannon divergence, though it is reported to show some tendency of mode seeking (Theis et al., 2015), it also effectively ameliorates the problem as it consistently outperforms the exclusive KL. Hence, we suggest that ParPI with W-distance could be the generally recommended setting. When the dimension of action is not large, JSD could also show good performance.

**Langevin Dynamics** We examine the choice of the step length $K$ and step size $\epsilon$ of MCMC steps. We observe that the sample that have higher Q-value could be fetched along with the MCMC steps,

Table 1: Results for D4RL benchmarks. Each number is the normalized score computed as (score - random policy score) / (expert policy score - random policy score) of the policy at the last iteration of training, averaged over 5 random seeds, ± standard deviation. Results of MOPO (Yu et al., 2020), BEAR (Wu et al., 2019) and CQL (Kumar et al., 2020) are reported from their respective papers. Remaining results are taken from the D4RL white-paper (Fu et al., 2020).

| Offline Types | Tasks | ParPI$_{WD}$+BRAC | ParPI$_{WD}$+MOPO | MOPO | CQL | BEAR | BRACv | BC |
|---|---|---|---|---|---|---|---|---|
| random | Walker2d | **9.9±3.2** | **15.2±8.2** | 13.6±2.6 | 7 | 7.3 | 1.9 | 1.6 |
| | HalfCheetah | **31.5±1.3** | **36.2±1.5** | 35.4±2.5 | 35.4 | 25.1 | 31.2 | 2.1 |
| | Hopper | 10.2±4.7 | 11.6±0.8 | 11.7±0.4 | 10.8 | 11.4 | 12.2 | 9.8 |
| medium | Walker2d | 79.1±7.2 | 39.4±13.2 | 17.8±19.3 | 79.2 | 59.1 | **81.1** | 6.6 |
| | HalfCheetah | **49.6±2.3** | 45.8±6.2 | 42.3±1.6 | 44.4 | 41.7 | 46.3 | 36.1 |
| | Hopper | **89.2±9.5** | 64.1±9.3 | 28.0±12.4 | 58 | 52.1 | 31.1 | 29.0 |
| medium-replay | Walker2d | **41.8±7.9** | 38.8±12.6 | 39.0±9.6 | 26.7 | 19.2 | 0.9 | 11.3 |
| | HalfCheetah | **39.9±1.7** | **54.8±3.1** | 53.1±2.0 | 46.2 | 38.6 | 47.7 | 38.4 |
| | Hopper | 42.1±8.1 | 64.5±27.1 | **67.5±24.7** | 48.6 | 33.7 | 0.6 | 11.8 |
| medium-expert | Walker2d | **105.6±3.4** | 91.2±11.2 | 44.6±12.9 | 98.7 | 40.1 | 81.6 | 6.4 |
| | HalfCheetah | 93.9±12.2 | **104.1±8.9** | 63.3±38.0 | 62.4 | 53.4 | 41.9 | 35.8 |
| | Hopper | **135.2±8.1** | 78.4±18.1 | 23.7±6.0 | 111 | 96.3 | 0.8 | 111.9 |
| expert | Walker2d | 108.0±3.1 | 99.4±10.1 | - | **153.9** | 106.1 | 0 | 125.7 |
| | HalfCheetah | **151.4±6.7** | 109.4±7.1 | - | 104.8 | 108.2 | -1.1 | 107 |
| | Hopper | **128.8±2.3** | 84.1±14.4 | - | 109.9 | 110.3 | 3.7 | 109 |

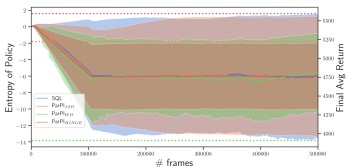
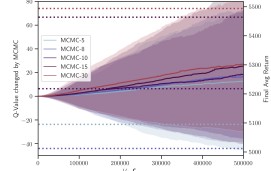
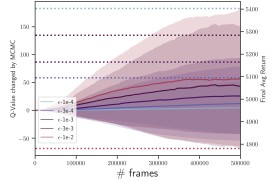

(a) Entropy of learnt policy      (b) The hyper-parameter of Langevin Dynamic analysis.

Figure 3: Study of component of ParPI on *HalfCheetah-v3*. Every experiment in ablation study was conducted using three different seeds.

and a longer MCMC run is or larger step size is, the sample has higher Q value as illustrated by Figure 3.

**Training Time Efficiency** To better understand the trade-off of ParPI on performance and training efficiency, we conduct a case study on walk clock time in Ant as shown in Table. 2. Here seconds to return indicates how much time is needed to gain the corresponding level of re-

Table 2: Wall-clock Time Comparison on Ant

| Seconds to Return | 1000 | 2000 | 3000 |
|---|---|---|---|
| ParPI$_{WD}$+SAC | 9463 | 11548 | 14544 |
| SAC | 5953 | 10006 | 12323 |

turns. ParPI would take slight more time as it takes additional time for MCMC steps. Moreover, the wall-clock time is usually not the main obstacle comparing to the environment time, i.e., time spent during interaction with the environment, in online settings and sample efficiency for offline settings.

## 5 CONCLUSIONS AND DISCUSSION

We devise a novel algorithms framework for model-free reinforcement learning, ParPI , a particle-based discrepancy/metric minimization framework for policy improvement, which can leverage the full potential of stochastic policy by enable the broad family of divergence and discrepancy, such as $f$-divergences and IPM-based metric. Our experiments on both online and offline settings show that the baseline algorithms benefit from the ParPI in that they achieve state-of-the-art performance with less hyper-parameter tuning efforts. ParPI accumulates many desirable properties: robustness, stochasticity, and sample efficiency. Furthermore, ParPI exhibits a stabilized training and has been shown to be time and sample efficient as compared to state-of-the-art approaches. Moreover, the ParPI is model invariant, allowing it adapt to different RL task settings.

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

## A  PARTICLE IMPROVEMENT WITH LANGEVIN DYNAMICS

As in the practical implementation, only the finite-step Langevin dynamics could be conducted. Fortunately, the property of MCMC guarantees the improvement of the particle distribution. With the $p$ denotes the unique stationary distribution, *e.g.*, the distribution implied by the $Q$-function in SQL and SAC. $q_t$ and $q_{t-1}$ refer to the distribution which is implicitly implied by an initial distribution and $t$ or $t-1$ steps Langevin dynamics, the following monotonic property is satisfied:

$$D_{KL}(q_t||p) \leq D_{KL}(q_{t-1}||p). \tag{10}$$

And $q_t$ converges to the stationary distribution $p$ as $t \to \infty$. The above proposition is the direct result of the following lemma, we also provide the proof here for the completeness.

**Lemma 1.** *Cover & Thomas (2012) Let $q$ and $r$ be two distributions for $z_0$. Let $q_t$ and $r_t$ be the corresponded distributions of state $z_t$ at time $t$, induced by the transition kernel $\mathcal{K}$. Then $D_{KL}[q_t||r_t] \geq D_{KL}[q_{t+1}||r_{t+1}]$ for all $t \geq 0$.*

*Proof.*

$$
\begin{aligned}
D_{KL}[q_t||r_t] &= \mathbb{E}_{q_t}\left[\log\frac{q_t(z_t)}{r_t(z_t)}\right] \\
&= \mathbb{E}_{q_t(z_t)\mathcal{K}(z_{t+1}|z_t)}\left[\log\frac{q_t(z_t)\mathcal{K}(z_{t+1}|z_t)}{r_t(z_t)\mathcal{K}(z_{t+1}|z_t)}\right] \\
&= \mathbb{E}_{q_{t+1}(z_{t+1})q_{t+1}(z_t|z_{t+1})}\left[\log\frac{q_{t+1}(z_{t+1})q(z_t|z_{t+1})}{r_{t+1}(z_{t+1})r(z_t|z_{t+1})}\right] \\
&= D_{KL}[q_{t+1}||r_{t+1}] + \mathbb{E}_{q_{t+1}}D_{KL}[q_{t+1}(z_t|z_{t+1})||r_{t+1}(z_t|z_{t+1})].
\end{aligned}
$$

$\square$

## B  POLICY IMPROVEMENT THEOREM FOR PARPI

Given the objective function as the expected discounted sum of rewards, the policy improvement theorem corresponds to how policies can be improved monotonically. Similar theorems have been derived under the maximum entropy framework, *i.e.*, SAC Haarnoja et al. (2018) and SQL Haarnoja et al. (2017). In this section, we shows that optimization of the policy with ParPI does not hurt the monotonic property.

### B.1  PARPI$_{SQL}$

In the SQL-version of ParPI, considering the original policy improvement theorem of SQL:

**Theorem 1.** *Haarnoja et al. (2017) Given a policy $\pi$, define a new policy $\tilde{\pi}$ as*

$$\tilde{\pi}(\cdot|\mathbf{s}) \propto \exp\left(Q_{soft}^\pi(\mathbf{s},\cdot)\right), \quad \forall \mathbf{s}$$

*Assume that throughout our computation, $Q$ is bounded and $\int \exp(Q(\mathbf{s},\mathbf{a}))d\mathbf{a}$ is bounded for any $s$ (for both $\pi$ and $\tilde{\pi}$). Then we have $Q_{soft}^{\tilde{\pi}}(\mathbf{s},\mathbf{a}) \geq Q_{soft}^\pi(\mathbf{s},\mathbf{a})\forall\mathbf{s},\mathbf{a}$.*

It should be noticed that there is no constraint on the policy optimization in Theorem. 1, hence we could directly replace the original reverse KL minimization with ParPI without hurting the policy improvement theorem.

### B.2  PARPI$_{SAC}$

In the situation of ParPI$_{SAC}$, we need to generalize the previous objective:

$$J_\pi(\phi) = \mathbb{E}_{\mathbf{s}_t \sim \mathcal{D}}\left[D_{KL}\left(\pi_\phi(\cdot|\mathbf{s}_t)\|\frac{\exp\left(Q_\theta(\mathbf{s}_t,\cdot)\right)}{Z_\theta(\mathbf{s}_t)}\right)\right]$$

to

$$J_\pi(\phi) = \mathbb{E}_{\mathbf{s}_t \sim \mathcal{D}}\left[D\left(\pi_\phi(\cdot|\mathbf{s}_t)\|\frac{\exp\left(Q_\theta(\mathbf{s}_t,\cdot)\right)}{Z_\theta(\mathbf{s}_t)}\right)\right]$$

To make it clear, we follow the proof provided in Haarnoja et al. (2018):

**Lemma 2.** *Let $\pi_{\text{old}} \in \Pi$ and let $\pi_{\text{new}}$ be the optimizer of the minimization problem defined in $J_\pi(\phi) := D(\rho(s)\pi_\phi(a|s), \rho(s)p(a|s))$. Then $Q^{\pi_{\text{new}}}(s,a) \geq Q^{\pi_{\text{old}}}(s,a)$ for all $(s,a) \in \mathcal{S} \times \mathcal{A}$ with $|\mathcal{A}| < \infty$*

*Proof.* Let $\pi_{\text{old}} \in \Pi$ and let $Q^{\pi_{\text{old}}}$ and $V^{\pi_{\text{old}}}$ be the corresponding soft state-action value and soft state value, and let $\pi_{\text{new}}$ be defined as

$$\pi_{\text{new}}\left(\cdot|\mathbf{s}_t\right) = \arg\min_{\pi' \in \Pi} \mathrm{D}_{\mathrm{KL}}\left(\pi'\left(\cdot|\mathbf{s}_t\right) \,\|\, \exp\left(Q^{\pi_{\text{old}}}\left(\mathbf{s}_t, \cdot\right) - \log Z^{\pi_{\text{old}}}\left(\mathbf{s}_t\right)\right)\right)$$

$$= \arg\min_{\pi' \in \Pi} J_{\pi_{\text{old}}}\left(\pi'\left(\cdot|\mathbf{s}_t\right)\right)$$

Note that $J_{\pi_{\text{old}}}\left(\pi_{\text{new}}\left(\cdot|\mathbf{s}_t\right)\right) \leq J_{\pi_{\text{old}}}\left(\pi_{\text{old}}\left(\cdot|\mathbf{s}_t\right)\right)$, since we can always choose $\pi_{\text{new}} = \pi_{\text{old}} \in \Pi$. Hence

$$\mathbb{E}_{\mathbf{a}_t \sim \pi_{\text{new}}}\left[\log \pi_{\text{new}}\left(\mathbf{a}_t|\mathbf{s}_t\right) - Q^{\pi_{\text{old}}}\left(\mathbf{s}_t, \mathbf{a}_t\right) + \log Z^{\pi_{\text{old}}}\left(\mathbf{s}_t\right)\right] \leq$$
$$\mathbb{E}_{\mathbf{a}_t \sim \pi_{\text{old}}}\left[\log \pi_{\text{old}}\left(\mathbf{a}_t|\mathbf{s}_t\right) - Q^{\pi_{\text{old}}}\left(\mathbf{s}_t, \mathbf{a}_t\right) + \log Z^{\pi_{\text{old}}}\left(\mathbf{s}_t\right)\right] \tag{11}$$

Applying soft Bellman Equation, we could get the policy improvement for SAC. $\square$

Note in ParPI$_{\text{SAC}}$, we could only need to guarantee the condition in Eq. 11 is also satisfied. This directly follows the monotonic property of Langevin dynamics as shown in Eq.10.

## C  DERIVATION OF SQUASH CORRECTION

To constrain the action space in a finite interval, the `tanh` is applied on the samples from the raw output $u$. And we also conduct the Langevin dynamics on $u$ space. The change of variable formula indicates the following equation is satisfied:

$$\log p(\mathbf{a}|\mathbf{s}) = \log p(\mathbf{u}|\mathbf{s}) \left|\det\left(\frac{\mathrm{d}\mathbf{a}}{\mathrm{d}\mathbf{u}}\right)\right|^{-1} = \log p(\mathbf{u}|\mathbf{s}) - \sum_{i=1}^{D} \log\left(1 - \tanh^2\left(u_i\right)\right)$$

The stationary distribution on the $a$ space is $exp(Q(s,a)/\alpha)$, the corresponding density function on the $u$ space satisfying that:

$$\log p(u|s) = \exp(Q(s, \tanh(u)\alpha) + 2\sum_{i=1}^{D} \log(1 - (\tanh(u_i)))$$

We take derivative and get the score function as:

$$\nabla_u \log p(u|s) = \nabla_u(Q(s, \tanh(u))/\alpha + 2\sum_{i=1}^{D} \nabla_u \log(1 - (\tanh(u_i)))$$

## D  DISCUSSION ON REMARK 2

The langevin dynamic is a special case of wasserstein gradient flow (Liu et al., 2019). At step $k+1$, the particle simulation is to solve the following problem:

$$\mu_{k+1}^{(h)} = \arg\min_{\mu} \mathrm{KL}(\mu\|p(\mathbf{x})) + \frac{W_2^2\left(\mu, \mu_k^{(h)}\right)}{2h}$$

Here $\mu_k$ denotes the sampled distribution after $k$-th step and $p(x)$ is the target distribution, *i.e.* $\frac{\pi_\beta(a|s)\exp(Q(s,a)/\alpha))}{Z}$ in our case. With limited steps of langevin dynamics, according to the triangle inequality of wasserstein distance we have:

$$\sum_{i}^{k} C_i \geq \sum_{i=1}^{k} \frac{W_2^2\left(\mu_{k+1}^h, \mu_k^{(h)}\right)}{2h} \geq \frac{W_2^2\left(\mu_{k+1}^{(h)}, \mu_1^{(h)}\right)}{2h} \tag{12}$$

The above inequality is to say that if each step the wasserstein distance could be bounded (by $C_i$), then the wasserstein distance between the final distribution and the original distribution is also bounded. And we replace $\mu_1^{(h)}$ with $\pi_\beta$ and $\mu_{k+1}^{(h)}$ with $\pi_{\phi_k}$, we get the $W_2^2\left(\pi_\beta, \pi_{\phi_k}\right)$ is constrained. Note that we optimize the wasserstein distance between $\pi_\phi$ and $\pi_{\phi_k}$ explicitly, and the wasserstein distance $W_2^2\left(\pi_\phi, \pi_{\phi_k}\right)$ is also constrained. Applying the triangle inequality agian, we get the

bounded condition on $W_2^2(\pi_\phi, \pi_\beta) \leq W_2^2(\pi_\phi, \pi_{\phi_k}) + W_2^2(\pi_\beta, \pi_{\phi_k})$. Integrating the $W_2^2(\pi_\phi, \pi_\beta)$ and $W_2^2(\pi_\phi, \pi_{\phi_k})$ into Eq. 12 with Lagrangian multiplier $\alpha$ and $\gamma$, we then get the Remark 2.

## E  CONVERGENCE ANALYSIS OF PARPI

The convergence property of ParPI is highly correlated with the particle updates in Eq. 4. Note that Jordan et al. (1998) indicates that with unlimited samples and infinitely small step size, the sample result could approach some stationary distribution $e^{U(\mathbf{x})}$. Such property provide a good theoretical intuition for convergence analysis on ParPI :

**Proposition 1.** *With unbiased gradient estimation on the $\nabla_\phi W_2^2(\pi_\phi, \pi_{sampled})$, where $\pi_{sampled}$ indicates the empirical distribution acquired from the langevin dynamics. If the sample size $M \to \infty$ and step size $\alpha \to 0$, $\pi_\phi$ in ParPI would converge to the global minimum $\frac{\pi_\beta(a|s)\exp(Q(s,a)/\alpha))}{Z}$.*

Note the Proposition 1 is based on the fact the $\mathrm{KL}(\cdot\|\cdot)$ is convex and we leave the detailed derivation in the Appendix.

To start with, we introduce the following lemma:

**Lemma 3.** *(Jordan et al., 1998) Assume that $\log p(\mathbf{x}) \leq C_1$ is infinitely differentiable, and $\|\nabla \log p(\mathbf{x})\| \leq C_2(1 + C_1 - \log p(\mathbf{x}))(\forall \mathbf{x})$ for some constants $\{C_1, C_2\}$. Let $T = hK$, $\mu_0 \triangleq q_0(\mathbf{x})$, and $\left\{\mu_k^{(h)}\right\}_{k=1}^K$ be the solution of the functional optimization problem:$\mu_{k+1}^{(h)} = \arg\min_\mu \mathrm{KL}(\mu\|p(\mathbf{x})) + \frac{W_2^2\left(\mu, \mu_k^{(h)}\right)}{2h}$, which are restricted to the space with finite second-order moments. Then i) the problem is convex;and ii) $\mu_K^{(h)}$ converges to $\mu_T$ in the limit of $h \to 0$, i.e., $\lim_{h \to 0} \mu_K^{(h)} = \mu_T$, where $\mu_T$ is the solution of Fokker-Planck (FP) equation: $\partial_\tau \mu_\tau = \nabla \cdot \left(-\mu_\tau \nabla U + \nabla \cdot \left(\mu_\tau \sigma \sigma^\top\right)\right)$ at $T$.*

Note the stationary distribution of the FP Equation is proportional to $e^{U(x))}$, Lemma 3 shows that $\lim_{k \to \infty, h \to 0} \mu_k^{(h)} = \frac{1}{Z}e^U$. In our case the stationary distribution refers to $\frac{\pi_\beta(a|s)\exp(Q(s,a)/\alpha))}{Z}$. Thus Lemma 3 suggests that with sample size $M \to \infty$ and step size $\alpha \to 0$, $\pi_\phi$ would converge to the global minimum $\frac{\pi_\beta(a|s)\exp(Q(s,a)/\alpha))}{Z}$.

## F  DISCUSSION ON MATCHING THE JOINT STATE-ACTION DISTRIBUTION

Note that ultimate goal of learning stochastic policy could be formulated as matching the conditional distribution, i.e., for every value of the state: $\min_\theta D\left(q_\theta(\cdot \mid s)\|p(\cdot \mid s)\right), \forall s$. While in this situation, the optimization problems need to be solved independently in different states. For example, if we want to optimize JSD or WD, then we will need many different discriminators or critics for different states. The corresponding computational complexity is unacceptable. To make the training tractable, we "amortized" the optimization problem on different states into a joint matching problem. Put it in another way, using the joint distribution is obvious when considering the analogy to supervised learning: the task is to match $f_\theta(x)$ to the corresponding target $y$ for every $x$, but a joint distribution is used in the objective: $E_{p(x,y)}\left[D\left(f_\theta(x), y\right)\right]$. The analogy is achieved by replacing the model $f_\theta(x)$ with $q_\theta(\cdot \mid s)$ and the target $y$ with $p(\cdot \mid s)$.

## G  EXPERIMENT DETAILS

All algorithms are implemented in Tensorflow (Abadi et al., 2016) and use a distributed implementation powered by Ray (Moritz et al., 2018). Following the implementation of TD3 (Fujimoto et al., 2018), we use two Q-value functions parameterized by a two-layer feed-forward network to fend off overestimation. A discriminator model sharing the same architecture is introduced for the divergence minimization purpose. Besides, we do not introduce any reward shaping for all tasks. When collecting rollouts for evaluations, we simply take the action selected by the policy at every state for every 1000 updates.

Policy, Q-value function & Discriminator architectures for both baselines and our algorithms:

| $s \in \mathbb{R}^s, a \in \mathbb{R}^a, z \in \mathbb{R}^a \sim \mathcal{N}(0, I)$ |
| :---: |
| Affine Transformation |
| Dense layer 256, ReLU |
| Dense layer 256, ReLU |
| Dense layer 256, Tanh |

Table 3: Policy

| $s \in \mathbb{R}^s, a \in \mathbb{R}^a$ |
| :---: |
| Dense layer 256, ReLU |
| Dense layer 256, ReLU |
| Dense layer 256, ReLU |
| dense $\rightarrow 1$ |

Table 4: (Double)Q Function

| $s \in \mathbb{R}^s, a \in \mathbb{R}^a$ |
| :---: |
| Affine Transformation |
| Dense layer 256, ReLU |
| Dense layer 256, ReLU |
| Dense layer 256, Linear |

Table 5: Discriminator

Table 6: Hyper-parameter Settings of ParPI

| Parameter | Policy | Q-function | Discriminator |
| :--- | :--- | :--- | :--- |
| optimizer | Adam | Adam | Adam |
| Learning rate | $3 \cdot 10^{-4}$ | $3 \cdot 10^{-4}$ | $3 \cdot 10^{-4}$ |
| discount($\gamma$) | 0.99 | n/a | n/a |
| replay buffer size | $10^6$ | n/a | n/a |
| entropy target | $-\dim(\mathcal{A})$ | n/a | n/a |
| MCMC steps | n/a | n/a | 5 |
| MCMC step size | n/a | n/a | $3 \cdot 10^{-4}$ |
| gradient steps | 1 | 1 | 1 |
| target update interval | 1 | 1 | 1 |

## H  OFFLINE ALGORITHM DETAILS

For completeness, we list the detailed algorithms for ParPI +MOPO in Algorithm 2. We use rollout length 5 for all tasks, and the same penalty coefficients reported by Yu et al. (2020).

---

**Algorithm 2** ParPI on top of MOPO

---

1: **Input:** reward penalty coefficient $\lambda$ rollout horizon $h$, rollout batch size $b$.
2: Train on batch data $\mathcal{D}_{\text{env}}$ an ensemble of $N$ probabilistic dynamics $\{\hat{T}^i(s', r|s, a) = \mathcal{N}(\mu^i(s, a), \sigma^i(s, a))\}_{i=1}^{N}$.
3: Initialize policy $\pi$ and empty replay buffer $\mathcal{D}_{\text{model}} \leftarrow \varnothing$.
4: **for** epoch $1, 2, \ldots$ **do**
5:     **for** $1, 2, \ldots, b$ (in parallel) **do**
6:         Sample state $s_1$ from $\mathcal{D}_{\text{env}}$ for the initialization of the rollout.
7:         **for** $j = 1, 2, \ldots, h$ **do**
8:             Sample an action $a_j \sim \pi(s_j)$.
9:             Randomly pick dynamics $\hat{T}$ from $\{\hat{T}^i\}_{i=1}^{N}$ and sample $s_{j+1}, r_j \sim \hat{T}(s_j, a_j)$.
10:            Compute $\tilde{r}_j = r_j - \lambda \max_{i=1}^{N} \|\Sigma^i(s_j, a_j)\|_{\text{F}}$.
11:            Add sample $(s_j, a_j, \tilde{r}_j, s_{j+1})$ to $\mathcal{D}_{\text{model}}$.
12:         **end for**
13:     **end for**
14:     Drawing samples from $\mathcal{D}_{\text{env}} \cup \mathcal{D}_{\text{model}}$, use **ParPI** to update $\pi$.
15: **end for**

---

For the ParPI +BRAC, following the original implementation (Wu et al., 2019; Fu, 2020), we using dual form for the behavior policy regulation(with KL) and value penalty with fixed $\alpha$. Then we use ParPi for both Value-function and Q-function update. We using (256,256) fully connected network as the critic in the minimax objective. We also adapt the gradient penalty in KL dual training. we using Adam for all optimizers.

