# OpenReview forum: "Particle Based Stochastic Policy Optimization"
_ICLR.cc/2022/Conference — ICLR 2022 Submitted_

### Official Review · Reviewer_Uu6f · 2021-11-02

**Correctness:** 2
**Technical Novelty And Significance:** 2
**Empirical Novelty And Significance:** Not applicable
**Recommendation:** 3
**Confidence:** 3

**Main Review:**

I have fundamental concerns about the paper.

- There are many typos and grammatical errors.
- Very little notation is defined and there is inconsistent use of the notation that is defined, making it difficult to understand and evaluate technical correctness and significance.
- The description of the algorithm is very convoluted and sparse. A large number of approximations are made, so it is not possible to infer it solely from the objective. Furthermore, the “policy updates” are never actually formally stated. A figure is included but it is not helpful in my opinion. I think this paper would greatly benefit from a general algorithm box early on that clearly explains the procedure (in contrast to Alg 1).
- There are statements of theorems and propositions but no proofs.
- The technical novelty is not clear. Although the particle-based idea could be new, many papers have considered general divergences and distances previously but there is little discussion of these. Minor point: the related work is abruptly in the middle of the paper. It would be helpful if the authors can clearly outline the differences between this work and prior work that has considered various forms of regularization and divergences.
- There are a number of claims throughout the paper stated as facts that are either very confusing or arguably unsubstantiated. Example: “Particularly in offline RL, the deterministic policy lacks the structure to manage risk towards uncertainty dynamics.” What structure does it lack? What are “uncertainty dynamics.” The data being offline does not change the fact that the optimal policy for the MDP is still deterministic (for nearly all MDPs of interest).
- The logic of the motivation is very hard to follow. It is continuously said that deterministic optimal policies are undesirable. Yet, if the policy is optimal, why does it matter if it is deterministic? Unless one is trying to also satisfy other desiderata, I’m not sure what the problem is. “Exploration” is cited as a reason, but this only matters during learning, not when the optimal policy is found. “Robustness” is cited but no robust problem is formulated. I would appreciate it if the authors can clarify this by formalizing mathematically the other desiderata that they would like the optimal policy to satisfy besides maximizing the reward. It appears that this is attempted in the objective in 2.1 with the entropy term, but it is not explained or proven that entropy alone satisfies these additional objectives beyond reward maximization.
- Furthermore, it is not clear why we should expect that a different discrepancy measure in the implementation should lead to differences in the solution to the objective stated in Sec 2. I suspect (correct me if I am wrong) that the authors would like these properties *during training* and not necessarily as properties of the optimal policy.
- One of the core problems that the paper tries to address is the fact that reverse KL is mode-seeking. However, this is never explained in the paper. Why is it mode-seeking? What effect does this have on the objective or the regularized objective? What properties of the other discrepancy measures suggest that they could solve this problem? I do not think it is sufficient to cite prior works that have also made this comment off-handedly. Instead, I think this is an important claim that should be understood via proof or experiments.

Specific comments and questions:
- Framework 1: I am confused about what purpose this serves. Was there supposed to be a theorem statement here?
- The “policy update” stated below Equation (1) is not explained/defined.
- D[ . || . ] notation is used inconsistently.
- Sec 2.1: V^* is never defined. Is this the actual optimal value function or the one for the regularized reward problem? These look like the normal Bellman equations, not the soft ones or regularized ones that are commonly used in the literature.
- Sec 2 on minimizing f-divergences:
    - What is the difference between q and q_\phi? What is g_\phi? Equation (2) is stated but seemingly never used. What is the purpose of this?
    - “Thus minimizing D˜ F well serves as minimizing DF” I think formal justification of this is necessary. Minimizing a lower bound does not mean you also minimize the upper bound in general.
    - It seems like many approximations are made in this paragraph but it is not clear what is lost by making these approximations or how close the approximations are to the truth.
- Sec 3:
    - V, V_{\psi}, and V_{\bar{\psi}} are not defined.
    - Equations (5) and (6) appear to be statements of residuals. What are the updates?
    - The discussion of Remark 2 seems dubious. It is claimed that a particular minimizer is recovered, but the discussion attempts to lower bound certain values and claim the minimizer is the same. This requires a formal proof.
    - “The first two terms in Eq. 8 correspond to the distribution constraint and the support constraint” It is not obvious to me that either constraint is satisfied since the constraints look “soft” now that they are additive terms in the objective. Can this be elaborated?
    - Equation (9) is very hard to read. Additionally there is use of $\approx$ without further discussion of what is being approximated. Can this be clarified?
- The figures are very small and hard to read.


**Summary Of The Paper:**

This paper proposes a new method for policy optimization in reinforcement learning. Building off of prior connections to probabilistic inference, the authors propose a framework for optimizing policies using general divergences/distances by way of particle-based methods for sampling. Some theory is presented. Experiments show that the method greatly out-performs state-of-the-art algorithms in both online and offline settings.


**Summary Of The Review:**

I do not believe this paper is ready to be published. There are a number of fundamental problems with the presentation and technical results. It may be that there are interesting or novel ideas here, as the experiments would seem to suggest, but unfortunately I believe it would require significant revisions.

I have the following high-level suggestions to improve the paper: The algorithm and basic method should be formally stated early on in a clear and concise way. When an approximation is made, it would be helpful to state it explicitly and justify it (if possible). It would be helpful to remove confusing claims and limit unnecessary discussion of prior work.

---

### Official Review · Reviewer_MUM2 · 2021-11-02

**Correctness:** 3
**Technical Novelty And Significance:** 3
**Empirical Novelty And Significance:** 3
**Recommendation:** 3
**Confidence:** 4

**Main Review:**

Review of: PARTICLE BASED STOCHASTIC POLICY OPTIMIZATION

I found the paper very difficult to understand. This was true in the language (see minor comments), as well as the logic (examples follow), and some basic details (Figures are unreadable, captions incomplete, key details missing).

The logic is hard to follow. For example:
"Modeling policy distribution by joint state-action distribution within the exponential family has enabled flexibility in exploration and learning multi-modal policies and also involved the probabilistic perspective of deep reinforcement learning (RL). The connection between probabilistic inference and RL makes it possible to leverage the advancements of probabilistic optimization tools. However, recent efforts are limited to the minimization of reverse KL divergence which is confidence-seeking and may fade the merit of a stochastic policy."

It is unclear what the goal and contribution of the paper is. To connect probabilistic methods with RL? What does it meant to fade the merit of a stochastic policy? Also, what does it mean to provide more flexible property?

Is framework 1 meant to be a mathematical statement? Is remark 1 supposed to follow from the framework? The structure of the math is difficulty to follow.

There is a lot of explaining in words that would be much clearer if stated and derived mathematically.


Minor:
- Stochastic policy has been <-- policies have been?
- "By providing better distribution matching methods" Is this demonstrated?
- Figure 1 is much too small. Also, I didn't understand it.
- How does ParPI "help" update the policy?
- Figure to is much too small. It is essentially unreadable.
- Why choose reverse KL and JSD for the experiments?
- What does PARPI + SQL mean? (Similar questions for all of the notation in the legend for figure 2)
- "We follow the same schema as the D4RL paper (Fu et al., 2020) to calculate the normalized return" This is not acceptable. The paper should state clearly what was done.
- " Jensen-Shannon divergence, though it is reported to
show some tendency of mode seeking (Theis et al., 2015), it also effectively ameliorates the problem as it consistently outperforms the exclusive KL" What does this mean?


**Summary Of The Paper:**

The paper introduces a particle-based approach for model-free RL. Multiple experiments compare the performance of the approach to SQL and SAC in online and offline settings. Results suggest that the proposed particle based approach improves on the prior work and shows different divergence measures yield different performance.


**Summary Of The Review:**

The paper proposes an interesting approach, but the writing and presentation were incomplete / unclear.

---

### Official Review · Reviewer_Rvy4 · 2021-11-03

**Correctness:** 4
**Technical Novelty And Significance:** 3
**Empirical Novelty And Significance:** 3
**Recommendation:** 5
**Confidence:** 4

**Main Review:**

Strong points:

The paper proposes a very interesting idea to solve an important problem in stochastic policy optimization. In supervised learning, we all know optimizing the reverse KL is a bad idea as it's mode-seeking. However, in reinforcement learning, many algorithms considers reverse KL as it's easy to compute the policy gradient from on-policy sampled transitions. The particle-based sampling is a very interesting solution to mitigate this problem. The proposed method is empirically verified on both online and offline RL tasks.

Weak points:

My concern is that this paper misses some important previous works that try to solve the same problem. See [1,2] for example. To compute the gradient of KL in policy optimization, one needs samples from the softmax policy defined by Q. The idea to implement this is to use self-normalized importance sampling. In fact, the methods proposed in [2] can exactly to be used for Eq 7.
I think it is important to discuss and compare these previous methods as they consider the exactly same problem of the paper.

Other problems:

In the online RL experiments, how is sampling from $\rho_\pi$ implemented? Does it need extra on-policy sampled trajectories, or just sample a transition from the replay buffer?


[1] Nachum, O., Norouzi, M., & Schuurmans, D. (2016). Improving policy gradient by exploring under-appreciated rewards. arXiv preprint arXiv:1611.09321.

[2] Mei, J., Xiao, C., Huang, R., Schuurmans, D., & Müller, M. (2019, August). On principled entropy exploration in policy optimization. In Proceedings of the 28th International Joint Conference on Artificial Intelligence (pp. 3130-3136).


**Summary Of The Paper:**

This paper proposes a particle-based sampling method for stochastic policy optimization in reinforcement learning. The main observation is that many policy optimization algorithms use reverse KL as the objective to optimize policies, which is mode-seeking and could be problematic in practice. To mitigate this problem, the authors propose to use particle-based sampling such that one can directly use KL for policy optimization.

**Summary Of The Review:**

I recommend rejecting this paper as important baselines are missed.

---

### Official Review · Reviewer_jv1X · 2021-11-05

**Correctness:** 4
**Technical Novelty And Significance:** 2
**Empirical Novelty And Significance:** 2
**Recommendation:** 5
**Confidence:** 3

**Main Review:**

I found that this paper contains some very interesting ideas, using Langevin dynamics to efficiently incorporate difference divergence frameworks is quite novel. However I found the presentation of the paper to be quite messy and insufficient, additional detailed comments are as follows:

- One major problem I found with this work is that the overall motivation seems unclear. While the authors demonstrated how their algorithm accommodates for f-divergence and Wasserstein distance. The authors failed to elaborate why using these divergence measures is desirable or where the performance boost they claim actually come from. While I appreciate the detail provided by the authors in the paper, the overall big picture not clear.
- In Algorithm 1, why did you sample the tuple (s, a, r) instead of (s, a, r, s')? If my understanding is correct, you are using the SAC or SQL value function updates (Equation 5 and 6) so you would need the next state s' to calculate the value targets.
- Can you clarify what "# frames" in the x-axis are referring to in Figure 2? If you are referring to the number of agent-environment interactions, then 1 millions samples is generally not considered sufficient for environments like Humanoid, at least 3 million or 10 million if you have the computational resources is generally recommended to get a good sense of the sample efficiency of the algorithm.
- Can you clarify the part on Langevin dynamics for Section 4.3? What are you referring to here when you say samples with a higher Q value can be fetched with larger step sizes or longer MCMC runs? Is this a bias issue? Also the captions in Figure 3 are unreadable, I suggest adding the full size figures to the appendix if there is not enough room in the main text.
- (Minor) Typo in paragraph before remark 2, specific 'choice' of divergence.
- (Minor) Was there a typo in Table 4 since at the beginning of the appendix you mentioned that you used a two-layered neural network for the Q functions?

**Summary Of The Paper:**

This work extends the popular RL as inference framework to accommodate for more general divergence measures such as the f-divergence and Wasserstein distance. The author proposed a particle-based optimization framework for learning stochastic policies where the policy is learned using samples generated via Langevin dynamics. The framework can be applied to both online and offline settings.

**Summary Of The Review:**

I think this line of work has great potential in its current state, I do not believe this paper is ready for publication.

---

### Decision · Program_Chairs · 2022-01-20

**Decision:**

Reject

**Comment:**

The manuscript extends the popular "RL as inference" framework with a generalized divergence minimization perspective. The authors observe that most policy optimization can be thought of as minimizing a reverse KL divergence, which has potentially undesirable mode-seeking properties. The authors propose a particle-based scheme wherein samples generated via Langevin dynamics are used for learning.

Several reviewers found the ideas presented interesting, and cited potential novelty and high potential for tackling an important problem. Unfortunately, all reviewers found major shortcomings, from presentation ("messy" presentation, lack of definition of notation and inconsistent use, issues around motivation and logical flow, vague and imprecise use of language, etc.). Several reviewers also had more fundamental criticisms, notably Uu6f who helpfully provided quite actionable feedback on the presentation. Unfortunately, discussion ended with the reviews: the authors offered no rebuttal or updates. The AC considers this a missed opportunity.

The AC concurs with, first and foremost, the concerns around presentation. The current state of the manuscript makes it difficult to parse apart the contribution being made, and in light of all 4 reviewers recommending rejection either strongly or weakly and with no rebuttals or responses put forth, I have no basis to recommend anything other than rejection.